# Buffering Capacity of Various Commercial and Homemade Foods in the Context of Gastric Canine Digestion

**DOI:** 10.3390/ani13233662

**Published:** 2023-11-27

**Authors:** Verena Stefani, Annegret Lucke, Qendrim Zebeli

**Affiliations:** 1Institute of Animal Nutrition and Functional Plant Compounds, Department for Farm Animals and Veterinary Public Health, University of Veterinary Medicine Vienna, Veterinaerplatz 1, 1210 Vienna, Austria; verena_stefani@yahoo.de (V.S.); alucke@nutrivet.uzh.ch (A.L.); 2Institute of Animal Nutrition and Dietetics, Vetsuisse Faculty, University of Zurich, 8057 Zurich, Switzerland

**Keywords:** canine, gastric digestion, buffering capacity, dogs

## Abstract

**Simple Summary:**

Knowing and predicting the buffering capacity of a food is of high importance in the context of gastric digestion and health. The aim of this study was to analyze the buffering capacity and the HCl amount needed to acidify a food, both as an indication of the acidity and gastric digestion of commercial and homemade dog foods in relation to their nutrient composition. The study developed prediction equations to estimate the buffering capacity using a set of 30 complete dog foods, each ten different types of commercial dry and wet dog food, and homemade dog food. To the best of our knowledge, this is the first study to evaluate canine food for buffering capacity.

**Abstract:**

The buffering capacity (BC) of food may act as a key regulatory parameter of canine gastric digestion by influencing the activity of gastric enzymes, the solubility of dietary ingredients, the gastric breakdown of food nutrients, and, subsequently, the absorption of nutrients. To analyse a possible effect of food on gastric pH, the BC of wet, dry, and homemade dog food was quantified via an acid titration method until a pH under 2 was achieved. Wet food had the highest BC; between dry and homemade food, there was no significant difference. Using multiple regression analyses, we were able to establish associations between the nutrient composition and the BC of the dog food. Crude protein content was the most important factor that influenced the BC and HCl use per gram of dry matter (DM) (*p* < 0.001), whereas the initial pH only tended to have an influence. The ash content also tended to affect the used HCl per gram of DM, and the DM content had a significant (*p* < 0.05) influence on the BC per gram of DM. The excessively high ash content found in wet food could be a risk factor for gastric dilatation–volvulus syndrome because it could lead to an insufficient pH drop in the stomach. Our data indicate large differences in the BC of typical dog food; so, estimating the BC using the equations developed herein could help to design individualized dog diets, in particular for dogs with health problems such as gastric hypoacidity, gastric reflux, or gastritis. However, more research about the influence of dog-food BC on gastric pH in vivo is needed.

## 1. Introduction

Buffering capacity (BC) is an important physiochemical property of food that expresses the resistance of the food to a change in pH with the addition of an acid or a base [1,2], thus having a direct influence on the absorption of hydrogen ions and, therefore, on the regulation of gastric juice pH [3]. In the context of gastric digestion, food BC may play a key regulatory role in digestion by influencing the gastric pH, especially by modulating the activity of gastric enzymes, the solubility of dietary ingredients, the gastric breakdown of food nutrients, and, therefore, the absorption of nutrients across the gastrointestinal tract [4]. Furthermore, the gastric pH acts as a barrier against foodborne pathogens [5] and may affect the gastrointestinal microbiota [6]. 

Therefore, the BC of food and dietary ingredients has frequently been the focus of research in the context of human gastric digestion [2,7,8,9]. For example, Salaün et al. (2005) analysed the BC of dairy products and it was found that the BC depends on the composition of minerals and proteins [7]. In Al-Dabbas et al. (2010), the BCs of legumes, almonds, lettuce stem, carob, liquorice root, and raw cow milk were measured, reporting a strong positive correlation of BC with protein and aspartic and glutamic acid contents [8]. In Mennah-Govela et al. (2020), the BCs of thirty commercially available foods were measured, identifying the protein content and initial pH of the food as the most important determiners of BC [2]. In more recent research by Ebert et al. (2021), ash, selected minerals, and amino acids, with a pKa in the range of foodstuffs, e.g., aspartic and glutamic acid, were detected as key influencing factors on overall food BC. In the study, wet texturized plant proteins were analysed for their BC, and the results were compared to the BC of pork meat [9]. 

To the authors’ knowledge, there are no studies dealing with the BC of canine food or dietary ingredients. This is surprising, taking into account the importance of BC not only for canine gastric digestion but also for canine health, such as gastric hyperacidity, gastroesophageal reflux, gastric hypoacidity or dilation–volvulus syndrome [10], food allergenicity [11], and dental health [12]. In addition, the design of oral canine drugs and pharmaceuticals requires knowledge of the food’s BC, too [13]. It has also to be mentioned that a dog’s diet is very complex, extending from the inclusion of meat, meat products, offal, and bones to dairy, eggs, and marine food products, and further to plant-based ingredients, such as grains, legumes, oilseeds, and vegetables, as well as various mineral supplements and feed additives [14]. These dietary ingredients are commonly fed as a commercially available complete food, either as dry kibbles or wet food, but also as a homemade diet. Therefore, both the ingredients and the manufacturing method may affect the BC of the dog food. Knowledge of the BC and pH of canine food may help in predicting the effects of the diet on digestion and gut health in dogs. The aim of this study was to analyze and compare the BCs of a variety of commercial dry and wet dog foods as well as homemade dog food in relation to their manufacturing method and nutrient composition. Our hypothesis was that, besides the protein content of the dog food, the manufacturing method will also have a major influence on its BC in the context of gastric digestion.

## 2. Materials and Methods

### 2.1. Dog Foods Used in the Experiment

In this experiment, a total of 30 complete dog-food samples were investigated, including randomly collected commercially available dry and canned foods as well as homemade dog food, with 10 different foods in each manufacturing method. The tested samples were dog foods prepared for healthy adult dogs (see Appendix A).

The commercial dog foods were purchased from different local supermarkets and pet stores to cover the variability of products with different producers, ingredients, and nutrient compositions. Dog food intended for particular nutritional purposes was not used. For compliance with nutritional standards, the producers of the commercial complete dog foods had to be members of the “Industrieverband Heimtier (IHV) e.V.” or the “Österreichische Heimtierfuttermittel Vereinigung (ÖHTV)”. Membership in one of these associations obliges the companies to produce their dog food according to the Fediaf Guidelines [15], both in terms of declaration and the current recommended nutrient levels for complete dog food. 

The homemade dog foods used common ingredients and were calculated to provide enough energy and nutrients for an adult, inactive dog using an Excel^®^ calculation program (“CarnivoreDiet” ©, A. Lucke, Vetmeduni, Vienna, Austria) according to nutrient requirements of dogs (National Research Council 2006). The meat used for the homemade diets was bought frozen and already minced in local pet stores. The vegetables were bought fresh in a local supermarket. The homemade dog-food diets were prepared and mixed on the same day as the analysis of the BC took place. The meat was cooked, and only the green tripe was used raw in some homemade diets. The amount of used ingredients to formulate the homemade food was calculated to meet the nutrient recommendations of an adult dog with a 7 kg body weight and weighed with a scale (ME4002^®^, Mettler Toledo, USA). Mineral supplements were added on top of the diet to ensure a well-balanced diet before mixing it together. 

### 2.2. Sample Preparation

For the sample preparation of BC measurement, 150 g of commercial dog food or the entire homemade dog food was mixed in a knife mill (Grindomix^®^ GM 200, Retsch, Haan, Germany). Canned dog food was homogenized five times for each for ten seconds at a speed of 5000 rotations per minute (rpm), dry dog food ten times for each for ten seconds at 5000 rpm, and homemade dog food three times for each for 30 s at 5000 rpm. Different mixing times were necessary to obtain comparable textures of the different dog food types. The texture of the dog food was not measured; the right texture was decided after visual judgment. 

Afterwards, 5 g of homogenized food and 15 g of deionized water (B30, Adrona, Riga, Latvia) were weighed (ME4002^®^, METTLER TOLEDO, Columbus, USA) and mixed in a 100 mL beaker. The final weight of each sample was 20 g. The rest of the homogenized food and an aliquot of at least 100 g of unhomogenized dry and wet dog food were frozen for further analyses. Homemade dog food was only frozen homogenised. 

For the measurement of the BC, a 0.16 M HCl solution was needed. Therefore, to a 0.1 mol/l ampoule for the preparation of Volumetric Solutions (ROTI^®^VOLUM, Carl Roth GmbH + Co. KG, Karlsruhe, Germany), distilled water was added until a total volume of 625 mL was reached. 

### 2.3. Measurement of the Buffering Capacity

The measurement of the BC was done by the acid titration method described previously [2]. In brief, aliquots of 0.5–2 mL of 0.16 M HCl were added to the sample until an endpoint of pH < 2 was reached [2]. The measurements of pH were carried out with a portable pH meter with a DHS electrode (pH 7^®^, Xs-Instruments, Carpi, Italy). The electrode was calibrated at room temperature using a standard buffer solution (Technical Buffer Solution, Mettler Toledo, Greifensee, Switzerland) and had to reach an accuracy from 95 to 105% before measuring the pH.

Before starting the titration, the pH values of the undiluted wet and homemade dog food were measured. Afterwards, the initial pH of the food samples consisting of 5 g food and 15 g deionized water of dry, wet, and homemade food was measured. To do so, the feed samples were stirred with a magnetic stirrer (MR Hei-Standard^®^, Heidolph Instruments, Schwabach, Germany) for a total of 5 min at 250 rpm. Then, 0.5 mL of 0.16 M HCl were added, and the samples were stirred for 30 s. After each addition of 0.5 mL of 0.16 M HCl, the pH value was measured. This procedure was repeated until the pH of the sample was below 2. To speed up the titration, the amount of 0.16 M HCl added per step was increased to 1 mL from a cumulative addition of 7 mL and to 2 mL from a cumulative addition of 30 mL of hydrochloric acid.

For quality control, a duplicate sample was measured for each dog food and each pH measurement was performed in triplicate.

### 2.4. Nutrient-Composition Analysis

All food samples were analysed for dry matter (DM), ash, crude protein (CP), ether extracts (EE), acid detergent fibre (ADF), and neutral detergent fibre (NDF) according to the guidelines of the Association of German Agricultural Analytic and Research Institutes [16]. The DM concentration was determined by oven-drying the samples at 103 °C for at least 4 h (method 3.1). The ash concentration was analyzed by combustion in a muffle furnace overnight at 580 °C (method 8.1). Ether extract was determined using the Soxhlet extraction system (method 5.1.2) and CP using the Kjeldahl method (method 4.1.1). A Fibretherm FT12 (Gerhardt GmbH and Co. KG, Königswinter, Germany) was used to obtain neutral detergent fibre assayed with a heat-stable α-amylase and expressed exclusive of residual ash (method 6.5.1). The nonfibre carbohydrates (NFC) were calculated as NFC = 100 − (NDF + CP + EE + ash). The homemade and wet-food samples had to be freeze-dried due to their high water content. For this purpose, the samples were deep-frozen overnight at minus 20 °C and then dried for 24 h under high-vacuum conditions (Lyovapor L-200, Büchi Labortechnik GmbH, Essen, Germany). After this process, the cooked and wet food samples were also dried overnight at 103 °C. 

### 2.5. Calculations

All calculations were done with Excel (Microsoft Excel, Microsoft Corporation^®^, Redmond, USA). First, the average of each triplicate pH measurement was calculated; this was also done with the duplicate of each sample. For further calculations, the average of each sample and its duplicate were used.

The calculation of the buffering capacity was based on acid titration curves [17].
(1)$$Total\;buffering\;capacity=\frac{total\;acid\;added}{∆pH}$$
(2)∆pH=initial pH−final pH

Based on the total buffering capacity, the buffering capacity per gram of DM of the dog food was calculated. Also, the HCl use per g of DM was calculated based on the total HCl use.

### 2.6. Statistical Analysis

For the statistical analysis of the data, the SAS (Version 9.4, SAS institute, Cary, NC, USA) was used. First, the data were tested for normality using the UNIVARIATE procedure of SAS. The homogeneity of the variances was tested graphically after checking the data for outliers using Cook’s D in SAS. Then, an analysis of variance (ANOVA) using the MIXED procedure of SAS was performed. The factor food type was defined as a fixed effect in the model statement and the independent food sample nested within the food type as a random effect. The Kenward–Roger method was used to approximate the degrees of freedom. A Tukey adjustment was applied to compare the means. 

A multiple regression analysis was performed with the backward elimination procedure to evaluate the influence of different dietary effects on HCl use per g of DM and BC per g of DM with PROC REG of SAS. The variance inflation factor (VIF) was computed to prevent multicollinearity among the predictors. The fitness of the model was tested using R^2^ and the root mean square error (RMSE). The *p* < 0.05 was considered significant and 0.05 *≤ p* < 0.10 as a tendency.

## 3. Results

### 3.1. Differences in Nutrient Composition among Three Food Types

The results of the ANOVA are shown in Table 1. There was no significant difference in the DM content between the wet (22.7 ± 1.39%) and homemade (26.4 ± 1.39%) food (*p* = 0.169). On a DM basis, the mean ash content of wet food (10.4 ± 0.46%) was significantly higher than in the dry (7.25 ± 0.49%) and homemade (6.30 ± 0.49%) food (*p* < 0.001). Wet food also had the highest estimated mean CP content (44.1 ± 0.49%) (*p* < 0.001). There was no significant difference in the CP content between dry (26.2 ± 1.44%) and homemade (29.1 ± 1.44%) food (*p* = 0.337). The EE content of the wet food (23.9 ± 2.30%) was significantly higher than in the dry food (12.3 ± 2.30%) (*p =* 0.004) but was not significantly higher than the EE content of the homemade food (18.8 ± 2.30%) (*p* = 0.273). Between the EE content of the homemade and dry food, there was no significant difference (*p* = 0.131). The ADF content of the homemade food (9.49 ± 1.23%) was significantly lower than the ADF content of the wet (17.3 ± 1.23%) (*p* < 0.001) and dry (14.6 ± 1.23%) food (*p* = 0.017). The NFC content of the wet food (5.88 ± 2.76%) was significantly lower than in the homemade (36.3 ± 2.76%) and dry (39.3 ± 2.76%) food (*p* < 0.001).

### 3.2. Buffering Capacity of Different Feed Types

The results of the ANOVA are shown in Table 2. The undiluted pH of homemade dog food was significantly lower than the undiluted pH of wet food (*p* < 0.001). There was a significant difference in the initial pH of all three food types (*p* < 0.001). Wet food had, with an estimated mean of 6.77 ± 0.08, the highest initial pH, followed by homemade food, with a mean pH of 6.31 ± 0.08. The lowest pH was dry food, with a mean of 5.62 ± 0.08. Dry food had, with 8.22 ± 0.34, compared to wet and homemade food, a significantly higher BC (*p* < 0.001). But if we look at the BC per gram of dry matter (BC/g DM), the wet food had, with 2.72 ± 0.14, a significantly higher BC/g of DM compared to the other food types (*p* < 0.001). Dry food had a mean BC/g of DM of 1.83, and homemade food had a mean BC/g of DM of 1.86. Matching the result of the BC/g DM, wet food had, with a mean of 13.20 mL, also a significantly higher used HCl per gram of dry matter (HCl/g DM) than the other food types (*p* < 0.001). Dry food had a mean use of HCl/g of DM of 6.69 and homemade food had a mean use of 8.15 mL. 

### 3.3. Associations between Nutrients and Variables of Buffer Capacity

Linear regression graphs were generated to evaluate the effect of nutrients on used HCl per gram of DM to reach pH < 2 as well as the measured BC of the food per gram of DM. Figure 1a–c shows linear and positive associations between the nutrient composition of the dog food and the used HCl per gram of DM. Accordingly, the data of Figure 1a revealed that 83% of the variance of used HCl/g of DM could be predicted with the percentage of crude protein in the DM of the dog food. According to this regression analysis, for each 1% CP in the diet, each g of DM food ingested would require 0.34 mL HCl to reach a pH < 2. Also, ash (R^2^ = 0.57) was a good predictor for the usage of HCl/g of DM, 1.16 mL HCl, for each g of DM of food, were needed for each 1% of ash to reach pH < 2 (Figure 1b). At the same time, the NFC (R^2^ = 0.60) content had a negative effect on the amount of HCl needed (Figure 1c). EE and ADF had no relevant effect on the used HCl per gram of DM. 

Figure 1d shows a polynomial regression of second degree between the used HCl per gram of DM and the initial pH. The coefficient of determination is 0.63. The initial pH correlated only positively with the amount of HCl for wet and homemade food, but less with dry food. The initial pH of dry does not seem to affect the amount of HCl per g of DM required to lower the pH < 2.

Figure 2a–c shows the linear regression between the nutrient composition of the dog food and the BC per gram of DM. Figure 2a shows that 72% of the variance of BC/g of DM could be predicted with the percentage of CP in the DM of dog food. Increasing the protein content by 1% led to an increase in the BC/g of DM of 0.06. Also, the ash (R^2^ = 0.60) led to an increase in the BC/g of DM of 0.2 per % increase of ash content (Figure 2b), whereas the NCF (R^2^ = 0.50) content in the DM of dog food had a negative effect on the BC/g DM (Figure 2c).

Figure 2d shows a polynomial regression of second degree between the used HCl per gram of DM and the initial pH. The coefficient of determination (R^2^) is 0.40. The initial pH correlated positively with the BC/g of DM of wet and homemade food, but less with dry food. The initial pH of the dry food does not seem to affect the BC/g of DM.

Figure 3 shows the linear regression between undiluted pH and initial pH, as compared with the ideal line (y = x). The graphic shows the effect of the addition of water to the dog food; the lower the pH of the food, the higher the effect of the addition of water, as indicated by the increasing distance of the regression line from the ideal line.

### 3.4. Mulitple Regression

A multiple regression was performed to evaluate the joint influence of and discriminate among different potential factors on HCl use per gram of DM and the BC per gram of DM of the food to reach a pH < 2. The results of the multiple regression are shown in Table 3 and Table 4.

The most significant positive influence on the HCl use per gram of DM is the percentage of crude protein (*p* = 0.0002) in the dog food, followed by the amount of crude ash (*p* = 0.0591), and the initial pH of the sample (*p* = 0.0769) (Table 3).

The most significant influence on the BC per gram of DM is the percentage of crude protein (*p* < 0.001) in the dog food, which affected positively the BC corrected per gram of food DM. In contrast, the DM content of the food (*p* = 0.0155) and its initial pH (*p* = 0.0850) both negatively affected the food’s BC corrected per gram of DM (Table 4).

## 4. Discussion

The aim of this study was to analyze the BC and the HCl amount needed to acidify the food, as an indication of the gastric acidity of commercial dog food and homemade dog food in relation to their nutrient compositions. Therefore, the buffering capacities of 30 complete dog foods, each of ten different types of commercial dry and wet dog food, and homemade dog food were measured and parameters that influenced the buffering capacity were evaluated. To the best of our knowledge, this is the first study to evaluate canine food for BC.

In our study, we could observe differences in initial pH, BC/g of DM, and used HCl/g of DM among the different food types. The lowest initial pH was dry food with a mean pH of 5.62, followed by homemade food, with a mean pH of 6.31. The highest pH was wet food, with a mean pH of 6.77. A possible explanation for the high initial pH in wet food could be the significantly higher CP content in wet compared to dry food. A study identified that feedstuffs with a high protein content have an initial pH around neutrality [18]. Homemade food had no significantly different protein content in the DM compared to dry food but had a significantly higher initial pH. Giger-Reverdin emphasized in her study that feedstuffs which retained water had lower initial pH values than feedstuffs of the same protein content but with a lower water-holding capacity [18]. The difference between these two different food types could be explained through the different DM content and preservatives or palatants, like phosphoric acid, citric acid, and mixed tocopherols that are added to dry food to enhance the palatability [19] or inhibit the growth of pathogens [20]. In this context, it is also interesting to mention that a low pH in dog food could possibly increase the risk of caries because it lowers the pH of the saliva [4,12]. 

There are also differences in the BC per gram of DM among the food types. The highest BC per gram of dry matter was wet food, with a mean of 2.72, followed by homemade food with a mean of 1.86. The lowest BC per gram of DM was dry food, with a mean of 1.83. It is no surprise that wet food had the highest BC per gram of DM because it also had the highest CP content. There was no significant difference in the BC per gram of DM between dry and homemade food; this was due to the similar CP content of these two food types.

The highest use of HCl per gram of DM was wet food, with a mean of 13.2 mL, followed by homemade food, with a mean of 8.15 mL. The lowest use of HCl per gram of DM was dry food with a mean of 6.69 mL. According to our results from the multiple regression, ash (*p* = 0.059) had a tendency to affect the HCl use and wet food had a significantly higher ash content than the other food types. This finding, in combination with the high CP content of the wet food, could explain the high HCl use. 

For all food types, there was a wide range in the values of BC for their own food type. This could be explained through their different nutrient composition, but also individual ingredients could have an influence on the initial pH, the BC per gram of DM and the HCl use per gram of DM. However, measuring the BC in commonly used ingredients in dog food would be necessary to prove the theory. In view of the results, the question arises whether the different food types have a different influence on gastric pH in dogs. There are no studies that measured the gastric pH of dogs during digestion and provided information on the BC of the used food. However, it is possible to calculate with our data the amount of gastric acid needed and the time to reach the gastric pH under two. It is estimated that dogs have postprandial gastric acid secretion around 1.5 mL per kg body mass per minute ([21], p. 42). For instance, if we take a 10 kg dog that is fed a meal of 70 g of food DM and take the mean acidity values of each food type, for wet food 924 mL gastric acid and 62 min are needed to reach a gastric pH under 2. In comparison, for homemade food, it is 570 mL gastric acid and 47 min, whereas for dry food 468 mL gastric acid and 32 min are needed to reach a gastric pH under 2. It is important that the calculated values cannot be applied one to one to the in vivo conditions in the stomach, since the gastric pH is influenced by many other factors too. However, this information could be important for dogs with gastric hypoacidity or gastritis to choose the right food, where a small amount of gastric acid is needed to reach the right pH levels in the stomach. The current dietary recommendation for dogs with gastritis is to feed restrictive CP content that covers the minimum requirements in dogs’ nutrition, in order to avoid increased gastric juice secretion ([21], pp. 274–275). For dogs with hypoacidity, it is recommended to feed diets based on easily digestible proteins and fat as energy sources ([21], pp. 274–275). 

Significant differences between the nutrient compositions could be found among the food types. Wet food had the highest CP content, with a mean of 44.1%, and was therefore significantly higher than the homemade (mean 29.1% DM) and dry food (mean 26.2% DM). The difference in CP content in dry and wet dog food could be explained through different demands on the manufacturing process of the food. A high level of starch is needed in dry food to maintain the durability of the kibble [22]; the production process of wet food does not require starch, so it mostly contains meat and animal byproducts [23]. The lower CP content of homemade food compared with wet food was due to the decision to feed rations with a lower protein content, enough to cover the dogs’ requirements in CP, but to cover the energy requirements mainly via carbohydrates and fats. 

The highest EE content was in the wet food, with a mean of 23.9% DM, followed by homemade food (mean 18.8% DM) and dry food (mean 12.3% DM). Surprisingly, the measured EE content in wet food was on average 13.3% lower and in dry food 7.9% lower than that declared by the manufacturers. Differences in the EE content could be caused by the different fat content of different parts of the carcass that were used for the dog food. 

Also, the different NFC content between the food types can be explained by the manufacturing process. The NFC provides information about the amount of cell ingredients, mostly starch and sugar, and pectin in a diet. This explains the high NFC of dry food, with a mean of 39.9% DM and homemade food, with a mean of 36.3% DM. 

In our study, wet food had a mean ash content of 10.4% DM, dry food 7.25% DM, and homemade food 6.3% DM. Furthermore, it is interesting to mention that, in our study, the measured ash content of wet food was on average 8.3% higher than that declared by the manufacturers. These results are quite similar to another study that measured the average ash content in dry food at 7.34% DM and in wet food at 10.1% DM [24]. Specifically, the high ash content in wet food could be a risk factor for gastric dilatation–volvulus syndrome, because it could lead to an insufficient pH drop in the stomach, which promotes the growth of gas-producing bacteria [25]. A possible explanation for the high ash content of wet food might be higher concentrations of Na and K [23]. NaCl is often added to wet food to increase the acceptance and palatability, and the addition of Na alginate, K alginate, or K carrageenin as gelling agents and thickeners [23]. Yet, as observed in this study, ash increases the BC of wet food significantly; therefore, our data suggest that the use of gelling agents in wet food needs additional evaluation. 

Regression analyses were used to search for various factors influencing BC/g DM and the use of HCl/g of DM. In our study, the content of CP was the most important factor that influenced the buffering capacity of dog food. In the multiple regression, CP had a very strong and positive significant influence on the BC/g of DM and used HCl/g of DM, both with a *p* < 0.001. Unfortunately, we do not have information about the exact protein composition of the tested food. Other studies have already shown that different protein sources, e. g. plant based or animal origin and different parts of the carcass, e. g. chicken skin or meat, have different BC values [9,26]. 

Also, other studies described protein as one of the main factors that affected the BC of food. For example, in Mennah-Govela et al., the BC of thirty commercially available commonly used food products that could be eaten as purchased, for example, milk, canned chicken, and tofu, were measured in her study and the protein content correlated with the total BC (R^2^ = 0.67) and the total acid added (R^2^ = 0.82) [2]. These results are similar to our results where the crude protein content correlated with the BC/g of DM (R^2^ = 0.72) and the HCl/g of DM (R^2^ = 0.83). Another study which measured the BC of commonly used feedstuff in ruminant diets concluded that the BC was high when the crude protein was high (>15%) and decreased as the protein content decreased [27]. In our study, wet food had the highest content of crude protein in DM, with 44.1% compared to dry food (26.2%) and homemade food (29.1%), and was therefore significantly higher. The BC/g of DM and the used HCl/g of DM were also significantly higher in wet food than in other food types. Further studies that analysed the BC of commonly used ingredients of pig and poultry feed came also to the conclusion that the protein content was an important factor [3,28].

The ash content of the food tended to affect the amount of used HCl/g of DM (*p* = 0.0591). A significant effect of ash content on the BC/g of DM was not proved after multiple regression. The linear regression between the BC/g of DM and ash R^2^ was 0.60 and between the used HCl/g of DM and R^2^ was 0.57. In Jasaitis et al. the BCs of fifty-two feeds, representing common ingredients used in ruminant diets, were analysed, and it was found that there was a significant correlation between BC and ash content (*p* < 0.001) [29]. The difference in the results of both studies could be due to a different cation–anion ratio in the ash content from dog food and ruminant feed. Anions also interact differently with different minerals; different compositions of minerals could also be a reason for the different results [7].

The initial pH only tended to have a negative influence on the BC/g of DM, with a *p* = 0.0850 and on the used HCl/g of DM a positive influence, with a *p* = 0.0769. This is in contrast to another study in which the initial pH had a negative significant influence on the BC (*p* < 0.05) but no significant influence on the total acid added (*p >* 0.05) [2]. 

Dry food had a mean DM content of 90.6%, which was a higher DM content than wet food (22.7%) and homemade food (26.4%). Also, the BC of dry food (8.22) was higher than the BC of wet food (3.02) and homemade food (2.40). These values could be explained through the measurement and calculation method of the BC. For the BC analysis, samples of 5 g from each food were used and the DM content of each food was not considered. However, if the BC was corrected for the DM, the DM content was considered, wet food had a mean of 2.72, a significantly higher BC/g of DM than dry food (1.83) and homemade food (1.86). This aspect shows why it is so important to compare the different feeds on a dry-matter basis; otherwise, the results would be misinterpreted. However, the multiple regression proved a negative significant influence (*p* < 0.05) of the DM content of the tested food on the BC/g of DM. This means that, the more water the food contains per gram of DM, the higher the BC/g of DM.

According to the research by Mennah-Govela et al., the particle size had a significant influence on food, with a protein content higher than 19% [2], and a CP content that is relevant to dog food. Accordingly, a smaller particle size resulted in a higher buffering capacity [2]. A possible effect could not be evaluated in our study, because the samples were mixed to obtain a comparable texture for the different dog feed types. However, the particle size could be important for the in vivo BC of the dog food. Dry food had, depending on the brand, a different croquette size. Also, in homemade diets, the particle size could play a role in the BC. For example, it could have an influence on the BC if minced meat is fed or meat sliced in pieces is fed. The possible influence of particle size on the BCs of different dog-feed types needs further research.

## 5. Conclusions

In our study, the CP concentration of the food was the most important factor that influenced the BC and HCl use, whereas the initial pH had only a weak significant influence. A possible influence of particle size of dog food on the BC needs further research. The BC can be estimated from the protein and ash content of the food, using the equations developed in this study. A high protein and ash content indicate a high BC of the food. However, more research about the influence of dog food BC on gastric pH in vivo is needed. In conclusion, the BC of food is a potentially interesting parameter for the future, as it can provide important information about the effect of food on digestion. And it can help to select the right diet according to the characteristic needs of a dog’s gastric digestion and health condition. 

## Figures and Tables

**Figure 1 animals-13-03662-f001:**
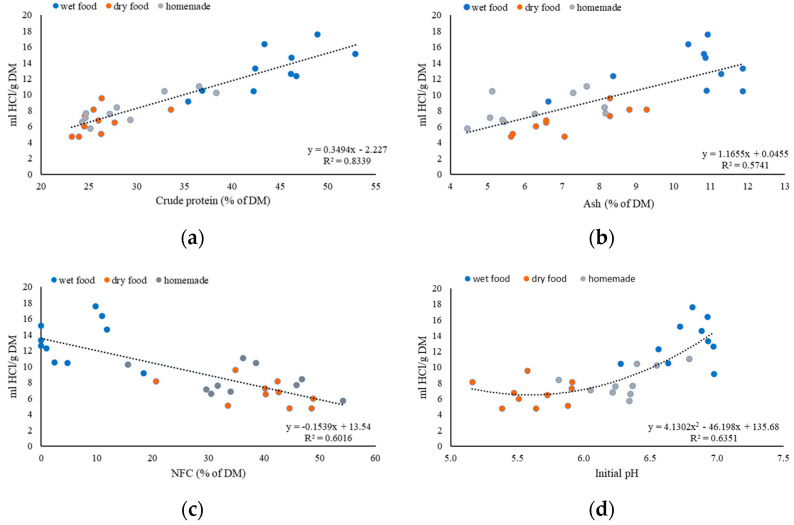
Effect of crude protein content, ash content, NFC content, and initial pH on the amounts of HCl per gram of DM wet (
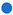
), dry (
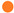
), and homemade (
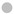
) food required to lower the pH < 2; (**a**) CP in % of DM; (**b**) ash in % of DM; (**c**) NFC in % of DM; and (**d**) initial pH.

**Figure 2 animals-13-03662-f002:**
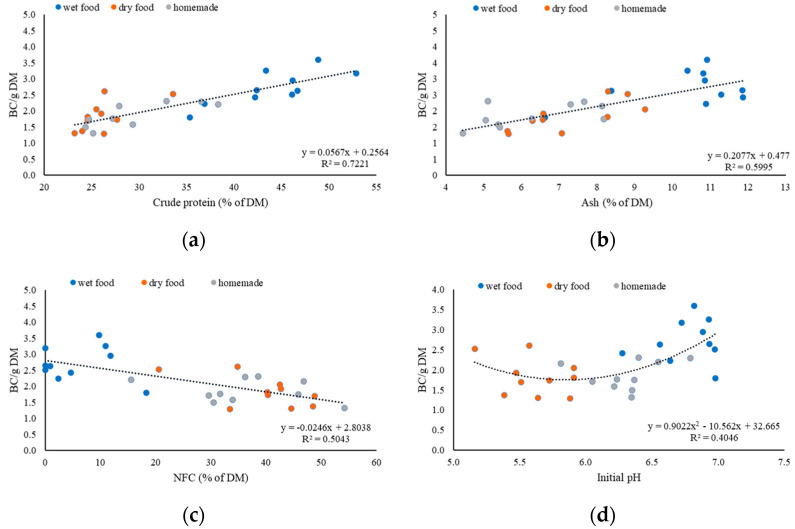
Effect of crude protein content, ash content, NFC content, and initial pH on buffering capacity per gram of DM wet (
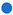
), dry (
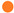
), and homemade (
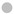
) food; (**a**) crude protein in % of DM; (**b**) ash in % of DM; (**c**) NFC in % of DM; (**d**) initial pH.

**Figure 3 animals-13-03662-f003:**
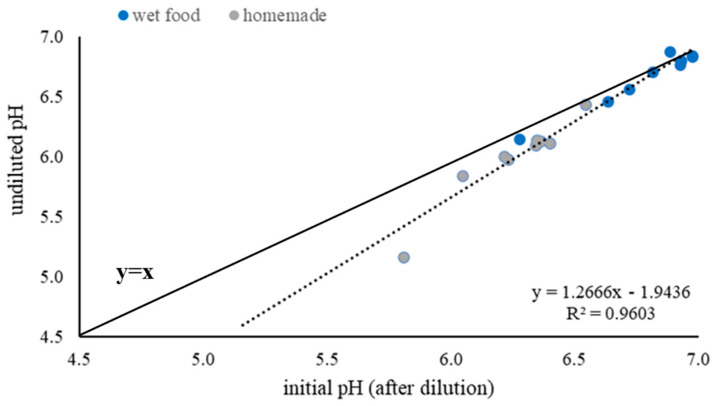
Relationship between undiluted pH of wet and homemade food, and its initial pH (after dilution with distilled water). The bolded line indicates the ideal line of this relationship (y = x).

**Table 1 animals-13-03662-t001:** Nutrient composition of different dog food types tested in this study [% DM unless otherwise stated].

Variable	Tested Food Type			SEM	*p* Value
	Wet Food	Homemade	Dry Food		
DM [% fresh]	22.7 ^b^	26.4 ^b^	90.6 ^a^	1.39	<0.001
Ash	10.4 ^a^	6.30 ^b^	7.25 ^b^	0.46	<0.001
CP	44.1 ^a^	29.1 ^b^	26.2 ^b^	1.44	<0.001
EE	23.9 ^a^	18.8 ^ab^	12.3 ^b^	2.30	0.005
ADF	17.3 ^a^	9.49 ^b^	14.6 ^a^	1.23	<0.001
NFC ^1^	5.88 ^b^	36.3 ^a^	39.6 ^a^	2.76	<0.001

Within a row, the means with different letters differ according to the Tukey test (*p* < 0.05).^1^ NFC = 100 − (NDF + CP + EE + ash).

**Table 2 animals-13-03662-t002:** Undiluted and initial pH, and buffering capacity of different dog food types.

Variable	Tested Food Type			SEM	*p* Value
	Wet Food	Homemade	Dry Food		
Undiluted pH ^1^	6.66 ^a^	5.98 ^b^	-	0.10	<0.001
Initial pH	6.77 ^a^	6.31 ^b^	5.62 ^c^	0.08	<0.001
Buffering capacity (BC)	3.02 ^b^	2.40 ^b^	8.22 ^a^	0.34	<0.001
Used HCl/g DM [mL]	13.2 ^a^	8.15 ^b^	6.69 ^b^	0.67	<0.001
BC/g DM	2.72 ^a^	1.86 ^b^	1.83 ^b^	0.14	<0.001

Within a row, the means with different letters differ according to the Tukey test (*p* < 0.05). ^1^ Undiluted pH is the pH of the food without any addition of water.

**Table 3 animals-13-03662-t003:** Effects of several dietary factors on the amount of HCl/g of DM, as evaluated by multiple regression analysis.

Variable	Parameter Estimate	Standard Error	*p*-Value	VIF	Adj. R^2^	Root MSE	Dependent Mean
Intercept	−8.90	3.49	0.017	0	0.85	1.34	9.34
Ash	0.33	0.17	0.059	2.29			
Crude protein	0.23	0.05	<0.001	3.79			
Initial pH	1.25	0.68	0.077	2.14			

VIF = variance inflation factor.

**Table 4 animals-13-03662-t004:** Effects of several dietary factors on BC/g of DM as evaluated by multiple regression analysis.

Variable	Parameter Estimate	Standard Error	*p*-Value	VIF	Adj. R^2^	Root MSE	Dependent Mean
Intercept	4.33	1.76	0.028	0	0.85	0.25	2.27
Crude protein	0.06	0.01	<0.001	2.32			
Initial pH	−0.50	0.27	0.085	2.38			
DM content	−0.04	0.01	0.016	1.61			

VIF = variance inflation factor.

## Data Availability

Data are available from the authors on reasonable request.

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
