# Peer review of "Buffering Capacity of Various Commercial and Homemade Foods in the Context of Gastric Canine Digestion"

_animals, 2023, doi:10.3390/ani13233662_

Round 1
Reviewer 1 Report
Comments and Suggestions for Authors
This is a well-written paper containing interesting results which merit publication. For the benefit of the reader, however a number of points need clarifying and certain statements require further justification. A few revisions are list below.
Q1: Ensure that p values need to be italicized in whole paper.
Q2: In page 5, “Crude protein” “Ether extracts” and “Acid detergent fibre” can be written as abbreviations.
Q3: The “Ash” in the article should be consistent.
Q4:. In page 5, there is no need for a period at the end of the title of the table.
Q5: In page 5, the letter marks in table 1 and table 2 should use the same standard.
Q6: In page 6, “Crude protein” can be written as their abbreviations.
Q7: In page 7, the table title and the table should be on the same page.
Q8: There is too much in the conclusion and it should be condensed.
Q9: In the result description section, adding p value will give better results
Q10: The last paragraph on page 9 can be combined with other paragraphs into one paragraph
Q11: The professional words that appear for the first time in the article should be written with full names and abbreviations, and abbreviations can be used when they appear again
Q12: The chemical formula for hydrochloric acid in the article should be an input error and should be “HCL”.
Q13: There is an additional “of” before the reference on line 47 and after the “and” on line 182.
Q14: There are errors in the reference citations of the author plus et al in the article, which should be in et al. Then insert the document and insert it in the correct format.
Q15: It is recommended to unify with or without “,” before “and” or “or” when connecting three consecutive items.
Q16: Missing ", "after reference 10 at line 61
Q17: The information from line 63 to line 71 lacks references to support It, especially the reference “It has also to be mentioned……”.
Q18: On line 70 it should be “help predicting of effects”.
Q19: when representing dog this species, it should be “dogs”.
Q20: The data difference “a b” in the table should be marked with “a” according to the largest average value, and those with a difference should be marked with “b” or “c”.
Q21: A space should be added before and after the “±” in the text.
Q22: The data in the article do not need to write out the average value and the corresponding data range in detail. The data range can be shown in the supplementary file if you want to provide it.
Q23: The algorithm for BC does not need to be explained again in the notes to Table 2, as it has been mentioned in Materials and Methods.
Q24: Line 232, “.” is missed before "also".
Q25: It should be expressed as “Figure 1a-c shows…, Figure 2a-c shows...” because the effect of the initial pH in picture d is not linear and is described piecewise.
Q26: What is the significance of measuring deviations between measured and declared nutrients?
Q27: On line 276 “buffering capacity” can be written as their abbreviations.
Q28: There's an extra “hat” on line 291.
Q29: Whether the insertion positions of references 17 and 18 can be merged.
Q30: The literature should not be inserted at line 302, because only the data results are described and no article is cited for discussion.
Q31: Would it not be more appropriate to merge the paragraph of the Discussion section of line 327~329 with the previous paragraph, plus references to the effect of food requirements on dogs with GI disorders? The paragraph of lines 347-350 should be merged with the next paragraph.
Q32: Parts of the article that are not important to the content of the article can be left out of the discussion, such as line 342-346, which has nothing to do with the buffering capacity of food.
Q33: The R2 on line 376~378 should be “R2”.
Q34: On line 391, an “and” is missed between “BC” and “ash content”.
Author Response
This is a well-written paper containing interesting results which merit publication. For the benefit of the reader, however a number of points need clarifying and certain statements require further justification. A few revisions are list below.
Authors: Thank you for your time and the constructive comments. We have done our best to take them into account during the revision process.
Q1: Ensure that p values need to be italicized in whole paper.
Authors: Thank you for your comment. The p-values were changed accordingly.
Q2: In page 5, “Crude protein” “Ether extracts” and “Acid detergent fibre” can be written as abbreviations.
Authors: Thank you for the hint. The wording was changed according to the suggestions.
Q3: The “Ash” in the article should be consistent.
Authors: Thank you for the correction. We changed the wording accordingly.
Q4:. In page 5, there is no need for a period at the end of the title of the table.
Authors: Thank you for the suggestion. We changed it accordingly.
Q5: In page 5, the letter marks in table 1 and table 2 should use the same standard.
Authors: Thank you for your remark. We adapted the letter marks as suggested.
Q6: In page 6, “Crude protein” can be written as their abbreviations.
Authors: Thank you for the comment. We changed the wording to the abbreviation.
Q7: In page 7, the table title and the table should be on the same page.
Authors: Thank you for your comment. The table is now on the same page.
Q8: There is too much in the conclusion and it should be condensed.
Authors: The conclusion was adapted and condensed as suggested.
Q9: In the result description section, adding p value will give better results
Authors: The p values were added everywhere in the result section as suggested.
Q10: The last paragraph on page 9 can be combined with other paragraphs into one paragraph
Authors:The paragraph is now combined with the previous one (L325).
Q11: The professional words that appear for the first time in the article should be written with full names and abbreviations, and abbreviations can be used when they appear again
Authors: Thank you for this comment. We rechecked the manuscript to ensure that abbreviations are used throughout.
Q12: The chemical formula for hydrochloric acid in the article should be an input error and should be “HCL”.
Authors: Thank you for this comment. We rechecked the correct abbreviation of hydrochloric acid and found that HCl is commonly used throughout the literature.
Q13: There is an additional “of” before the reference on line 47 and after the “and” on line 182.
Authors; Thank you for the comment. We changed the wording in line 47 accordingly and rechecked line 182. The additional “of” in line 182 was found on line 192 and removed.
Q14: There are errors in the reference citations of the author plus et al in the article, which should be in et al. Then insert the document and insert it in the correct format.
Authors: The wording in the article was changed to “in author et al.”.
Q15: It is recommended to unify with or without “,” before “and” or “or” when connecting three consecutive items.
Authors: We put a comma before “and” and “or” when three consecutive items where connected.
Q16: Missing ", "after reference 10 at line 61
Authors: We put a comma after reference 10 (L62).
Q17: The information from line 63 to line 71 lacks references to support It, especially the reference “It has also to be mentioned……”.
Authors: References were added (L64, L68)
Q18: On line 70 it should be “help predicting of effects”.
Authors: Thank you, we changed the wording (L71).
Q19: when representing dog this species, it should be “dogs”.
Authors: Thank you, the wording was changed in the text.
Q20: The data difference “a b” in the table should be marked with “a” according to the largest average value, and those with a difference should be marked with “b” or “c”.
Authors: Thank you for your correction. The data differences “a b” were changed according to your suggestion.
Q21: A space should be added before and after the “±” in the text.
Authors: A space was added before and after “±” in the text.
Q22: The data in the article do not need to write out the average value and the corresponding data range in detail. The data range can be shown in the supplementary file if you want to provide it.
Authors: Thank you for your comment. The data range was removed in the article.
Q23: The algorithm for BC does not need to be explained again in the notes to Table 2, as it has been mentioned in Materials and Methods.
Authors: The algorithm of BC in the notes to Table 2 was removed.
Q24: Line 232, “.” is missed before "also".
Authors: The missing “.” was added (L242)
Q25: It should be expressed as “Figure 1a-c shows…, Figure 2a-c shows...” because the effect of the initial pH in picture d is not linear and is described piecewise.
Authors: The wording was changed according to your suggestion.
Q26: What is the significance of measuring deviations between measured and declared nutrients?
Authors: The deviations were measured to see what extent the declared nutrients were correct. Since the data has no further relevance for the paper, the data was added to the supplementary material.
Q27: On line 276 “buffering capacity” can be written as their abbreviations.
Authors: Thank you for the comment. We changed it accordingly (L 296).
Q28: There's an extra “hat” on line 291.
Authors: Thank you for the comment. We changed the wording to “had” (L 311)
Q29: Whether the insertion positions of references 17 and 18 can be merged.
Authors: Thank you for the suggestion. References 17 and 18 (due to changes now reference 19 and 20) were merged as suggested (L306).
Q30: The literature should not be inserted at line 302, because only the data results are described and no article is cited for discussion.
Authors: We removed the literature (L313).
Q31: Would it not be more appropriate to merge the paragraph of the Discussion section of line 327~329 with the previous paragraph, plus references to the effect of food requirements on dogs with GI disorders? The paragraph of lines 347-350 should be merged with the next paragraph.
Authors: The paragraph was merged with the previous one (L 325). Information about food requirements was added in line 342 (“The current feeding recommendation for dogs with gastritis is to feed CP content, that covers the minimum requirements in dogs’ nutrition to avoid increased gastric juice secretion [22]. For dogs with hypoacidity, it is recommended to feed diets based on easy digestible proteins and fat as energy source [22].”). In our opinion, it is not appropriate to merge the two paragraphs (L350-353), as they do not fit together in terms of content.
Q32: Parts of the article that are not important to the content of the article can be left out of the discussion, such as line 342-346, which has nothing to do with the buffering capacity of food.
Authors: Thank you for the suggestion. We condensed this part of the discussion (L 369-370) accordingly.
Q33: The R2 on line 376~378 should be “R2”.
Authors: Thank you for this correction. We changed the spelling as suggested.
Q34: On line 391, an “and” is missed between “BC” and “ash content”.
Authors: Thank you for the commend, the wording was corrected as suggested (L 411).
Reviewer 2 Report
Comments and Suggestions for Authors
It is a quite original study that can lead to different recommandations in formulating canine diets in the future. Ash content is specifically a problem in many low quality diets.
Minor comments
-"in vivo" should appear in the title
-125. Description of pH measurements of Home-made and wet diets. What was the method for dry foods ?
-There is not enough informations about commercial diets (adult ? growing dogs ? high fiber.....) : how is the sample (n=30) representative of the products on the market ? (any PARNUT diet ? )
-The differences in measured vs declared ingredients (Table 5) is interesting but the question remain : " is the difference related to the content (large différences for high contents of nutrients , or the opposite ) or not ?
-338 : I do not agree with the author about "a low protein content" = 29.1 % DM in dogs ? So, this sentence would be re-written in a more nuanced and correct way.
On the whole, the discussion is well constructed.
Author Response
It is a quite original study that can lead to different recommandations in formulating canine diets in the future. Ash content is specifically a problem in many low quality diets.
Minor comments
-"in vivo" should appear in the title
Authors: Thank you for the comment, The study was conducted as an "in vitro" experiment, not as "in vivo".
-125. Description of pH measurements of Home-made and wet diets. What was the method for dry foods ?
Authors: Thank you for your comment, The description of the pH measurements were specified (L131, “the initial pH of the food sample consisting of 5 g food and 15 g deionized water of dry, wet, and homemade food was measured.”).
-There is not enough informations about commercial diets (adult ? growing dogs ? high fiber.....) : how is the sample (n=30) representative of the products on the market ? (any PARNUT diet ? )
Authors: We added additional information about the commercial diets in line 82 and 86 (“ The tested samples were dog foods for healthy adult dogs.The commercial dog foods were purchased from different local supermarkets and pet stores to cover variability of products with different producers, ingredients, and nutrient compositions. Dog food intended for Particular Nutritional purposes was not used.”)
-The differences in measured vs declared ingredients (Table 5) is interesting but the question remain : " is the difference related to the content (large différences for high contents of nutrients , or the opposite ) or not ?
Authors: Thank you for your comment. The deviations were measured to see what extent the declared nutrients were correct. Since the data has no further relevance for the paper, the data was added to the supplementary material.
-338 : I do not agree with the author about "a low protein content" = 29.1 % DM in dogs ? So, this sentence would be re-written in a more nuanced and correct way.
Authors: Thank you for your comment. The sentence was changed (L375-377, “The lower CP content of homemade food compared with wet food was due to the decision to feed rations with a lower protein content, enough to cover the dogs’ requirements in CP, but to cover the energy requirements mainly via carbohydrates and fats.”). We hope the sentence is now more appropriate.
On the whole, the discussion is well constructed.
Round 2
Reviewer 1 Report
Comments and Suggestions for Authors
I suggest to accpet in present form
Author Response
Thank you!